# Topological phase locking in stochastic oscillators

Michalis Chatzittofi [1], Ramin Golestanian [1,2] ✉ & Jaime Agudo-Canalejo [1,3] ✉

The dynamics of many nanoscale biological and synthetic systems such as enzymes and molecular motors are activated by thermal noise, and driven out-of-equilibrium by local energy dissipation. Because the energies dissipated in these systems are comparable to the thermal energy, one would generally expect their dynamics to be highly stochastic. Here, by studying a thermodynamically-consistent model of two coupled noise-activated oscillators, we show that this is not always the case. Thanks to a novel phenomenon that we term topological phase locking (TPL), the coupled dynamics become quasi-deterministic, resulting in a greatly enhanced average speed of the oscillators. TPL is characterized by the emergence of a band of periodic orbits that form a torus knot in phase space, along which the two oscillators advance in rational multiples of each other. The effectively conservative dynamics along this band coexists with the basin of attraction of the dissipative fixed point. We further show that TPL arises as a result of a complex, infinite hierarchy of global bifurcations. Our results have implications for understanding the dynamics of a wide range of systems, from biological enzymes and molecular motors to engineered nanoscale electronic, optical, or mechanical oscillators.

Enzymes and molecular motors are pivotal in catalyzing biochemical reactions and converting chemical energy into mechanical work[1,2]. By dissipating energy at the molecular scale, they play a crucial role in the maintenance of life's out-of-equilibrium dynamics. However, because the dynamics in these systems tend to involve noise-activated barrier-crossing processes with energy scales comparable to the thermal energy, $k_B T$, their dynamics tend to be highly stochastic. To be more reliable, biological systems have developed various strategies that trade off energy dissipation for increased precision[3–5], as exemplified by e.g. proofreading[6,7] or noise buffering strategies[8,9]. Thermal noise and stochasticity similarly play an important role for synthetic motors operating at the nanoscale[10].

One possible strategy for reducing stochasticity and in turn increasing reliability lies in many-body interactions, i.e., synchronization[11,12]. For example, in the case of the KaiABC circadian clock, the collective oscillations of many interacting KaiABC protein complexes are significantly more coherent than those of a single complex[13,14]. In the KaiABC system, interactions are "chemical," in the sense that they are mediated by monomer exchange among the complexes. However, because enzymes and molecular motors transduce chemical energy into motion, they also experience "physical", or mechanical, interactions with each other through the viscous medium in which they are embedded, see Fig. 1a–d. The viscous nature of the medium leads to interactions mediated by hydrodynamic friction, or dissipative interactions.

Due to the key role that the interplay between thermal fluctuations and nonequilibrium driving energies plays in these systems, one must be particularly careful when modeling their dynamics. Thermodynamic consistency, in particular the requirement that local detailed balance and a fluctuation-dissipation relation be satisfied, strongly constrains the form of the dynamics[15]. Recently, using a minimal thermodynamically-consistent model for two identical enzymes that

[1]Max Planck Institute for Dynamics and Self-Organization (MPI-DS), Göttingen, Germany. [2]Rudolf Peierls Centre for Theoretical Physics, University of Oxford, Oxford, UK. [3]Department of Physics and Astronomy, University College London, London, UK. ✉e-mail: ramin.golestanian@ds.mpg.de; j.agudo-canalejo@ucl.ac.uk

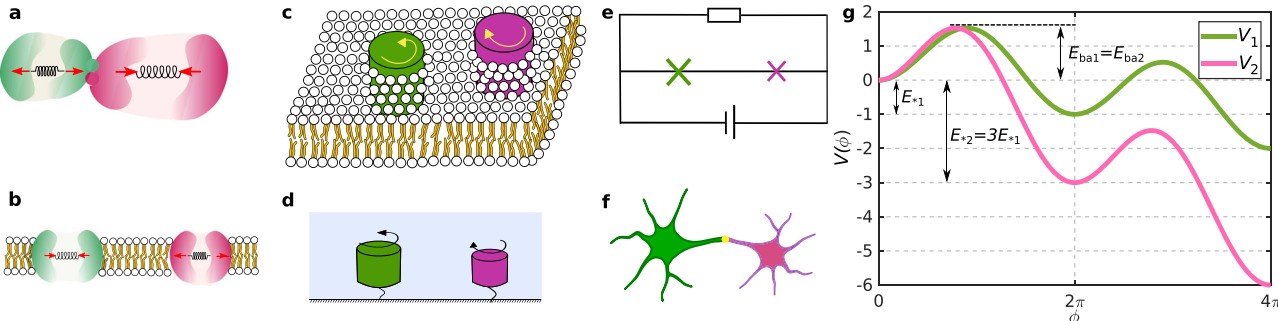

**Fig. 1 | Examples of coupled noise-activated oscillators. a** Two enzymes attached to each other forming an oligomeric complex, **b** Two membrane channels interacting with each other via the intervening viscous medium, **c** Two rotating inclusions in a membrane, **d** Two molecular rotors interacting hydrodynamically, **e** A circuit with two Josephson junctions, **f** Two excitable neurons interacting through a synapse. **g** The internal phase $\phi$ of each oscillator experiences a driving force represented by a tilted washboard potential, with a noise-activated oscillation corresponding to the phase advancing by the amount of $2\pi$ by crossing the energy barrier. In the case of enzymes, the potential represents the free energy of a (repeated) catalytic reaction.

are mechanically-coupled to each other and undergo conformational changes during their reaction cycle, we showed that the mechanochemical coupling in these systems can cause synchronization and enhanced reaction speeds[16]. A generalization of this model to arbitrary numbers of coupled identical noise-activated oscillators shows synchronization at low numbers of oscillators, and enhanced speeds independent of the number of oscillators[17]. Interestingly, the transition to the synchronized state in that model was shown to occur as a result of a global bifurcation in the underlying dynamical system, which transitions from purely dissipative, noise-activated dynamics (where all trajectories lead to the fixed point) to a mixture of dissipative and conservative dynamics (where some trajectories are periodic and avoid the fixed point) beyond a critical coupling strength[16,17]. This very intriguing bifurcation has also been reported in the context of coupled superconducting Josephson junctions[18].

While fascinating, the latter results have limited applicability, as they only concern coupled identical oscillators. Here, we study the dynamics of two nonidentical noise-activated oscillators that are governed by different free energy landscapes and nonequilibrium driving forces and are dissipatively coupled. Crucially, our model is generic enough that it may serve as a minimal model for a wide variety of cyclic nano-scale systems. Examples include dissimilar enzymes (see Fig. 1a, b)[19] or gating nano-pores[20], nano-scale rotary motors (see Fig. 1c, d) (either biological, such as ATP synthase[21], or synthetic, such as those recently made from DNA-origami[22–24]), circadian clocks[25,26], superconducting Josephson junction arrays (Fig. 1e)[18,27], firing neurons (Fig. 1f)[28,29], artificial systems like magnetic rotors[30], laser cavities[31,32], opto-mechanical devices[33,34], mechanical oscillators[35,36], or any other suitably-reduced description of an excitable system[37,38].

We find that, instead of a single bifurcation occurring with increasing coupling strength as for identical oscillators, nonidentical oscillators undergo an infinite number of bifurcations as the coupling is increased. The oscillators are generically phase-locked, such that noise activation leads to a finite number of oscillations for each oscillator, with a fixed ratio between them. Moreover, for sufficiently strong asymmetry in the nonequilibrium driving forces, a finite number of "resonant" modes emerges at specific values of the coupling strength. For these resonant modes, we find periodic trajectories that avoid the fixed point and maintain a fixed ratio between the number of steps advanced by each oscillator. To reach (or move away from) these resonant modes, an infinite ladder of bifurcations must be climbed (or descended). We find that the resonant modes correspond to very special topologies of the deterministic phase portraits of the system, defined on the torus, in which the phase space splits into a band of periodic orbits, which form torus knots[39] with a specific winding number. We thus refer to this novel phenomenon as topological phase

locking (TPL). In the stochastic dynamics, TPL results in a greatly enhanced average speed as well as giant diffusion[40], which together create strong signatures in the stochastic thermodynamics of the precision of the coupled oscillators[4,41].

## Results

### Dissipative coupling of noise-activated processes

We consider two processes, each defined by a phase $\phi_\alpha$ with $\alpha = 1, 2$, which evolve along two washboard potentials $V_\alpha(\phi_\alpha)$, see Fig. 1g. The key parameters of the potential are the height of the energy barrier $E_{ba\alpha}$, which determines the noise-activated dynamics, and the energy released per transition $E_{*\alpha}$, which acts as the nonequilibrium driving force. The two phases are coupled not through an interaction force or potential, but through the off-diagonal components of the mobility tensor that connects forces to velocities in the overdamped dynamics. That is, the phases evolve according to the following coupled Langevin equations

$$\dot{\phi}_\alpha = \sum_{\beta=1}^{2}\left[ M_{\alpha\beta}\left(-\frac{\partial V_\beta(\phi_\beta)}{\partial \phi_\beta}\right) + \sqrt{2k_B T}\,\Sigma_{\alpha\beta}\xi_\beta \right], \tag{1}$$

where $M_{\alpha\beta}$ is the mobility tensor, described below, $\Sigma_{\alpha\beta}$ is the principal square root of $M_{\alpha\beta}$ such that $M_{\alpha\beta} = \Sigma_{\alpha\gamma}\Sigma_{\beta\gamma}$, and $\xi_\alpha(t)$ is a Gaussian white noise satisfying $\langle \xi_\alpha(t)\rangle = 0$, $\langle \xi_\alpha(t)\xi_\beta(t')\rangle = \delta_{\alpha\beta}\delta(t - t')$. Moreover, $k_B$ is the Boltzmann constant and $T$ is the temperature of the medium, so that $k_B T$ is the thermal energy controlling the strength of thermal fluctuations. For non-thermal systems, $k_B T$ may be taken as the strength of the effective noise. For the dynamics to be thermodynamically consistent, the mobility tensor must be symmetric and positive definite[42,43]. We take the components of the mobility tensor to be $M_{11} = \mu_1$, $M_{22} = \mu_2$, and $M_{12} = M_{21} = \sqrt{\mu_1\mu_2}h$. Thus, the dimensionless parameter $h$ controls the strength of the coupling, and the condition of positive definiteness implies that it is constrained to the range $-1 < h < 1$. Through $\Sigma_{\alpha\beta}$, the mobility tensor also controls the form of the additive noise, so that the fluctuation-dissipation theorem is satisfied. This further implies that, independently of the strength of the coupling, the system is guaranteed to equilibrate to the Boltzmann distribution $P_{eq}(\phi_1, \phi_2) \propto \exp(-[V_1(\phi_1) + V_2(\phi_2)]/k_B T)$ when such an equilibrium is possible (e.g., in the absence of nonequilibrium driving forces, $E_{*1} = E_{*2} = 0$).

A coupling of the form in Eq. (1) arises naturally in processes that are coupled to each other through mechanical interactions at the nano- and microscale, as these are mediated by viscous, overdamped fields described by low Reynolds number hydrodynamics[43]. It represents a form of dissipative coupling, as it can be understood as arising from taking the overdamped limit of full Langevin dynamics in the

presence of a friction force on phase $\phi_\alpha$ going as $f_\alpha = -\sum_{\beta=1}^{2} Z_{\alpha\beta}\dot{\phi}_\beta$, where $Z \equiv M^{-1}$ is a friction tensor. As an example, we show in the Supplemental Methods how Eq. (1) can be derived from a microscopic model of two rotors that are hydrodynamically coupled (see Fig. 1c, d)[44]. In this case, the coupling $h$ is exactly constant, and its magnitude and sign are governed by the rotation rate and chirality of the rotors. In a similar way, one can derive coupled phase equations for enzymes that undergo conformational changes (Fig. 1a, b), where they reduce to exactly the same form but with a phase-dependent coupling constant $h(\phi_1, \phi_2)$ which moreover leads to multiplicative noise[16].

The potentials are chosen to be tilted washboard potentials of the form $V_\beta(\phi_\beta) = -F_\beta\phi_\beta - v_\beta\cos(\phi_\beta + \delta_\beta)$, where the shift $\delta_\beta = \arcsin(F_\beta/v_\beta)$ ensures that the minima of the potential are located at multiples of $2\pi$ and does not otherwise affect the phase dynamics. The maxima of the potential are located at $\phi_\beta^{max} \equiv \pi - \arcsin(F_\beta/v_\beta)$ (mod $2\pi$). The parameters $F_\beta$ and $v_\beta$ can be mapped to the energy barrier and the energy released per step (Fig. 1g) as $E_{ba\beta} = [2\sqrt{1 - (F_\beta/v_\beta)^2} - (F_\beta/v_\beta)(\pi - 2\delta_\beta)]v_\beta$ and $E_{*\beta} = 2\pi F_\beta$. In the following, except where noted, we focus on the case of equal self-mobilities $\mu_1 = \mu_2 = \mu$, equal energy barriers $E_{ba1} = E_{ba2} = E_{ba}$, and strongly driven dynamics $E_{*1} \gg E_{ba}$ (we fix $E_{ba}/E_{*1} = 3 \cdot 10^{-4}$). Choosing a mobility scale $\mu_0$ and an energy scale $E_0$, which together define a timescale $(\mu_0 E_0)^{-1}$, only three dimensionless parameters remain: $E_{*2}/E_{*1}$, which governs the asymmetry in the nonequilibrium driving of the two processes and we take to be $\geq 1$ (i.e., oscillator 2 is more strongly driven than oscillator 1); $h$, which defines the strength of the dissipative coupling; and $k_B T/E_{ba}$, which defines the strength of the noise. Note that, for $E_{*1} = E_{*2} = E_*$, the problem reduces to that of two identical oscillators, previously studied in ref. 16 as well as in ref. 17 for the general case of $N \geq 2$ identical oscillators.

## Stochastic trajectories

We briefly present the phenomenology observed in stochastic simulations of the equations of motion, Eq. (1), when the dissipative coupling is switched on (Fig. 2). In the absence of coupling, as expected, the trajectories are independent, and consist of single steps representing noise-activated crossings of the energy barriers in the potential, separated by long periods of time in which the phases are resting at the minima of the potential (see Fig. 1d). With sufficiently large positive coupling, on the other hand, we observe that when the system is pushed out of the resting state, both oscillators advance at the same time, and moreover, multiple steps occur as a result of a single fluctuation. This results in an overall enhanced average speed of the oscillators. In contrast to the synchronous steps previously observed for identical oscillators with $E_{*1} = E_{*2}$[16,17], nonidentical oscillators with $E_{*1} \neq E_{*2}$ do not appear to be synchronized, but there are signatures of

phase locking, where $\phi_1$ advances $n_1$ steps while $\phi_2$ advances $n_2$ steps with a reproducible ratio $n_1{:}n_2$, in this example 2:3.

Importantly, this behavior is apparent even at very low values of the noise. This suggests that, as in the case of identical oscillators[16,17], the phase locking phenomenology may be a consequence of bifurcations occurring in the underlying deterministic dynamical system.

## Finite phase locking

We start by analyzing the phase portraits in $(\phi_1, \phi_2)$ space corresponding to the deterministic part of Eq. (1). Because the dynamics are $2\pi$-periodic, this dynamical system is defined on the torus. Notice that the system always has four fixed points: a stable fixed point at $(0,0)$, corresponding to both oscillators being at a minimum of their potential energy; an unstable fixed point, at $(\phi_1^{max}, \phi_2^{max})$ when both at are a maximum; and two saddle points at $(\phi_1^{max}, 0)$ and $(0, \phi_2^{max})$, when one oscillator is at a minimum and the other at a maximum. Because of the structure of Eq. (1), the location and character of these fixed points is independent of the strength of the coupling. In particular, this means that local bifurcations (where fixed points split or merge and change character) are impossible. Any bifurcation in this system must be global, arising from a change in the topology of the network of heteroclinic and homoclinic orbits connecting these four fixed points[45].

Phase portraits for weak driving force asymmetry $E_{*2}/E_{*1} = 5$ and several values of the coupling $h$ are shown in Fig. 3. The labels $(m, n)$ are winding number pairs, describing how many times a trajectory starting in that region will wind around the torus along each dimension before reaching the stable fixed point. More precisely, if the torus is unwrapped and tiled onto the plane, a trajectory starting anywhere in the region $-\pi \leq \phi_1, \phi_2 < \pi$ is said to have winding number $(m, n)$ if it ends at the stable fixed point at $(\phi_1, \phi_2) = (2\pi m, 2\pi n)$. This convention allows us to associate winding numbers to trajectories even if they are not closed, and in turn to assign winding numbers to different sub-regions within the region $-\pi \leq \phi_1, \phi_2 < \pi$, and ultimately to classify different topologies of the phase portraits according to the winding number $(m, n)$ associated to the longest trajectory on that phase portrait. Thus, the phase portraits in Fig. 3a–d can be said to have topologies (1, 1), (1, 2), (2, 3), and (3, 4), respectively. In Fig. 3, every point of the phase portrait (except those at heteroclinic orbits, which connect the unstable fixed point to the saddle points) has been colored according to the Euclidean distance between the point in question and the fixed point (for an unwrapped torus) that a trajectory starting at that point would reach. Thus, yellow corresponds to longer trajectories toward the fixed point, whereas blue corresponds to shorter trajectories.

In the planar phase portraits (Fig. 3a–d), regions with different winding numbers appear to be separated from each other by the heteroclinic orbits. However, on the surface of the torus (Fig. 3e, f), one can appreciate that the region enclosed by the heteroclinic orbits is

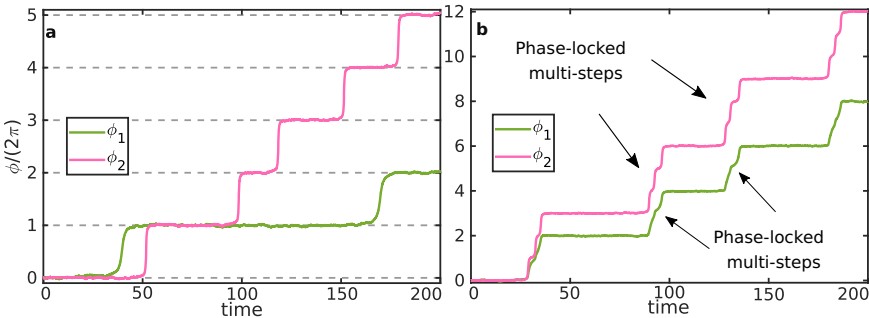

**Fig. 2 | Examples of nonidentical stochastic trajectories. a** In the absence of coupling, $h = 0$, only a few, independent single steps are observed. **b** With coupling, $h = 0.33$, a much larger overall number of steps is observed in the same time period,

and moreover, both phases move in tandem, in phase-locked, multi-step bursts. In both examples, we fixed asymmetry $E_{*2}/E_{*1} = 5$ and noise strength $k_B T/E_{ba} = 1$. Time is given in units of $(\mu v_1)^{-1}$.

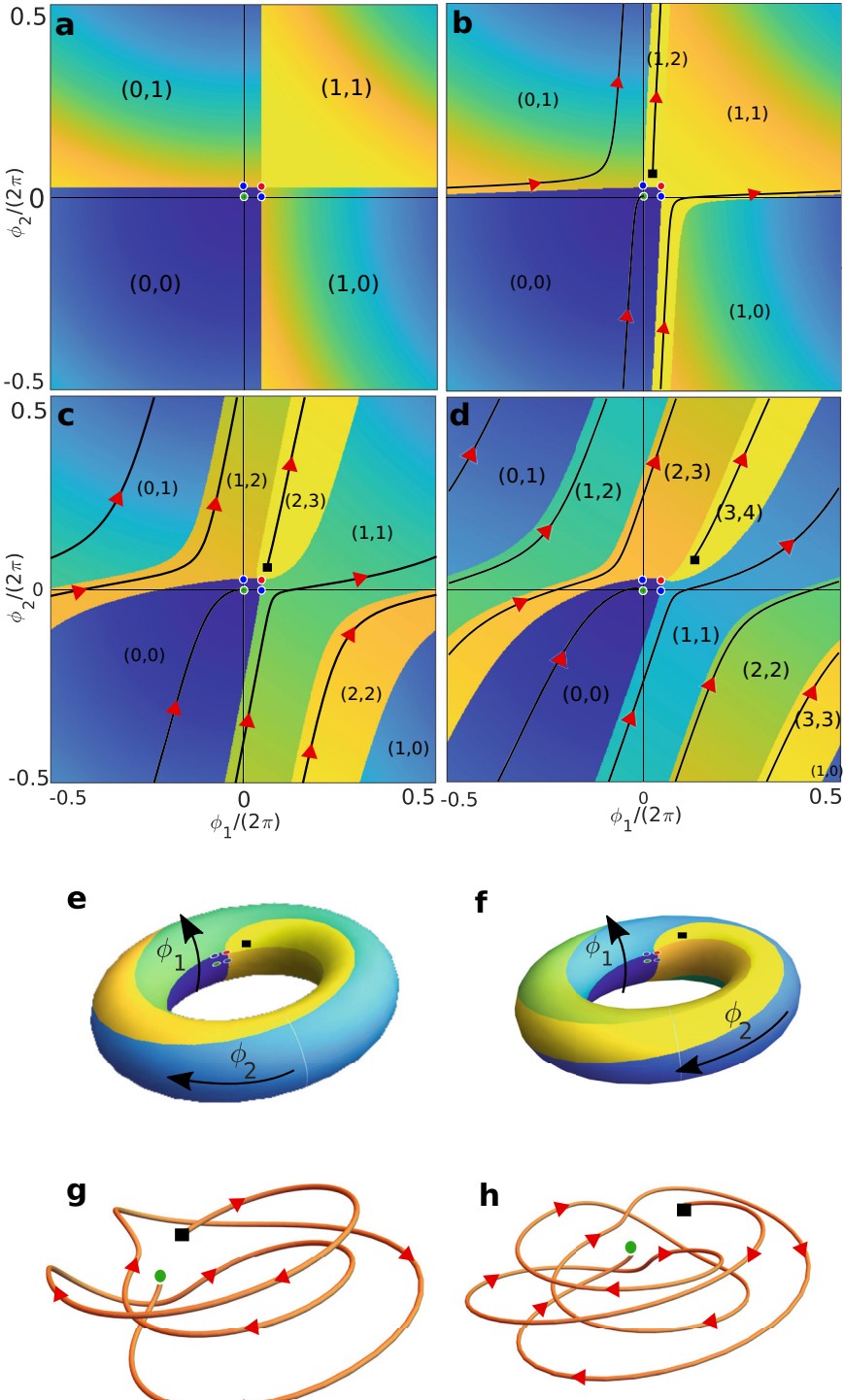

**Fig. 3 | Phase portraits of the deterministic dynamics for weak asymmetry.** **a** (1, 1) topology for $h = 0$. **b** (1, 2) topology for $h = 0.05$. **c** (2, 3) topology for $h = 0.19$. **d** (3, 4) topology for $h = 0.33$. In all panels the green, red, and blue circles, respectively, correspond to the stable, unstable, and saddle fixed points of the dynamics. An example trajectory, starting at the black square and finishing at the stable fixed point, is shown in **b**–**d**. The phase portraits in **c** and **d** are represented on the torus in **e**, **f**. The example trajectories in **c** and **d** are represented as three-dimensional trajectories around the torus in **g** and **h**. The colormap in **a** and **f** and the labels $(m, n)$ in **a**–**d** are explained in the text. The asymmetry was fixed as $E_2/E_1 = 5$.

still simply connected, covers the whole torus, and corresponds to the basin of attraction of the stable fixed point. With increasing coupling, we observe a series of global bifurcations in the heteroclinic network, which change the maximal winding numbers that are possible from e.g., (1, 1) in the absence of coupling (Fig. 3a) to (3, 4) for $h = 0.33$ (Fig. 3d). This higher winding implies that the basin of attraction becomes a narrower and narrower strip, which winds around the torus

an increasing number of times given by the highest winding number pair.

These bifurcations are responsible for the phenomenology observed in the stochastic simulations of Fig. 2, which we term finite phase locking. Indeed, let us take the phase portraits in Fig. 3a, d as an example. In the presence of fluctuations, a system initially located at the stable fixed point will typically be kicked by noise over either of the

saddle points. In the absence of coupling, Fig. 3a, this implies that the system enters either the (1, 0) or the (0, 1) basin, so that just one of the oscillators undergoes a single step. With coupling, Fig. 3d, the system instead enters the (2, 3) or the (1, 1) basin, resulting in a finite number of steps taken in tandem by the two enzymes. Note that, typically, one of the two saddle points will be more easily reachable and thus traversed much more frequently than the other[46,47]. It is also important to note that, although the maximal winding number pair in Fig. 3d is (3, 4), observing a (3, 4) transition in the stochastic system should be rare, as the system will typically escape the stable fixed point through one of the saddle points, and not through the unstable point. In this particular case, the stochastic simulations in Fig. 2b confirm that the $(0, \phi_2^{\max})$ saddle point is preferred, as all the stochastic transitions observed lead to a (2, 3) transition. The time-course of a (2, 3) stochastic transition is shown on top of the corresponding deterministic phase portrait in Supplementary Movie 1.

### Topological phase locking

For sufficiently strong driving force asymmetry, at specific values of the parameters belonging to a subset of codimension 1 in parameter space, we find phase portraits that are qualitatively different, see Fig. 4a, b. The topology of the heteroclinic network changes, resulting in the formation of two homoclinic orbits that connect each of the two saddle points to itself. As a consequence, the phase space becomes disconnected into two regions: the basin of attraction of the stable fixed point, and a band of closed, periodic orbits (in gray in Fig. 4a, b). We refer to this phenomenon as TPL.

Importantly, a nontrivial winding number pair can also be assigned to the running band region. In the particular example of Fig. 4, we observe that a periodic orbit (and, by extension, the running band region as a whole) winds two times along the $\phi_1$ direction and three times along the $\phi_2$ direction before closing in on itself, implying a winding number pair, which we denote as $(2, 3)_\infty$ in analogy with the notation for winding number pairs introduced above, where the $\infty$ subscript indicates that the trajectories are closed and never reach a fixed point.

An example of a closed, periodic trajectory within the running band is shown in Fig. 4a, with the three-dimensional view of its projection on a torus shown in Fig. 4c. It is interesting to note that the loop formed by the trajectory corresponds to a trefoil knot which, naturally,

belongs to the class of torus knots (knots that lie on the surface of a torus)[39].

TPL has very strong consequences in the stochastic dynamics. In the presence of fluctuations, a system initially located at the stable fixed point will now be kicked by noise over either of the saddle points and fall into the running band. The phases $\phi_1$ and $\phi_2$ will then advance deterministically, in the ratio given by the corresponding winding number pair, until a sufficiently strong fluctuation kicks the system out of the running band and back into the stable fixed point. The average speed of the oscillations can therefore be greatly enhanced by the presence of a running band. The time-course of a stochastic multi-step run in a $(2, 3)_\infty$ TPL state is shown on top of the corresponding deterministic phase portrait in Supplementary Movie 2.

### Phase-locking diagram

To understand how and where these different phase portrait topologies emerge in parameter space, as well as the global bifurcations that connect them, we scanned the parameter space as a function of driving force asymmetry $E_{*2}/E_{*1}$ and coupling strength $h$. The topologies of phase portraits with finite phase locking were identified by means of the highest winding number pair $(m, n)$, whereas those corresponding to TPL were identified using the winding number pair $(m, n)_\infty$ of their running band.

The resulting phase-locking diagram, shown in Fig. 5, demonstrates an incredibly rich structure of bifurcations in the system. Note that the colors in the diagram correspond to the logarithmic value of the second number $n$ in the winding number pair $(m, n)$, with blue corresponding to low numbers and red to high numbers. We find a variety of regions corresponding to phase portraits with finite phase locking with different winding numbers. Most interestingly, however, we observe a number of dark red branches or resonances at which the winding numbers very sharply peak as we vary $E_{*2}/E_{*1}$ and/or $h$ and cross through the resonance. At the very center of these resonances, in a lower-dimensional manifold of codimension 1, we find the phase portraits with TPL (TPL states).

To better understand the bifurcation structure, let us focus on the $(1, 2)_\infty$ TPL state, which is the first one to appear as the coupling $h$ is increased. Suppose we begin at the dot marked (2, 5) at the top-left of Fig. 5, which corresponds to finite phase locking. As we increase $h$, we first observe a bifurcation to (3, 7), i.e., the maximal winding numbers

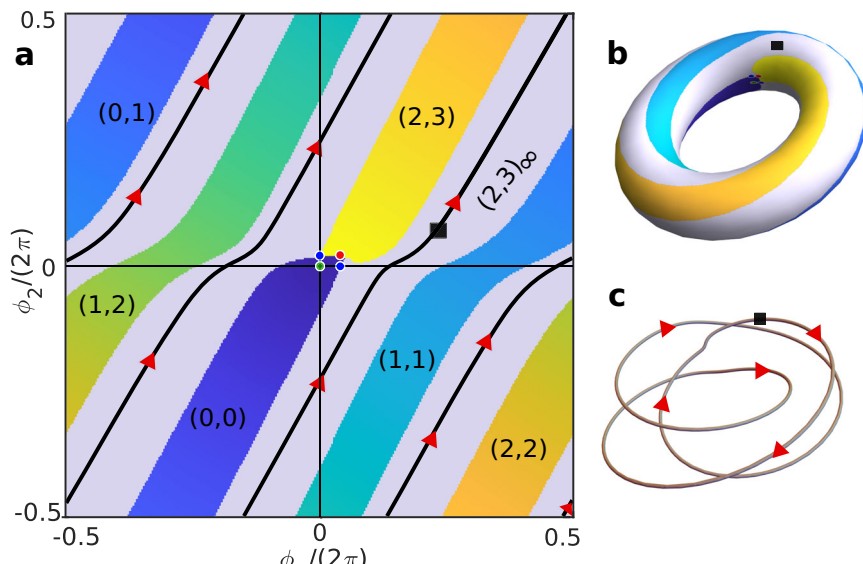

**Fig. 4 | $(2, 3)_\infty$ TPL state. a** Phase portrait for strong asymmetry $E_{*2}/E_{*1} = 19.15$ and $h = 0.51$. The running band is shown in gray. The black line crossing through the black square (shown as a reference point) is an example of a periodic orbit within the running band. **b** The $(2, 3)_\infty$ TPL state projected on a torus. **c** The black solid line in **a** is depicted as a three-dimensional loop around the torus, which forms a trefoil knot.

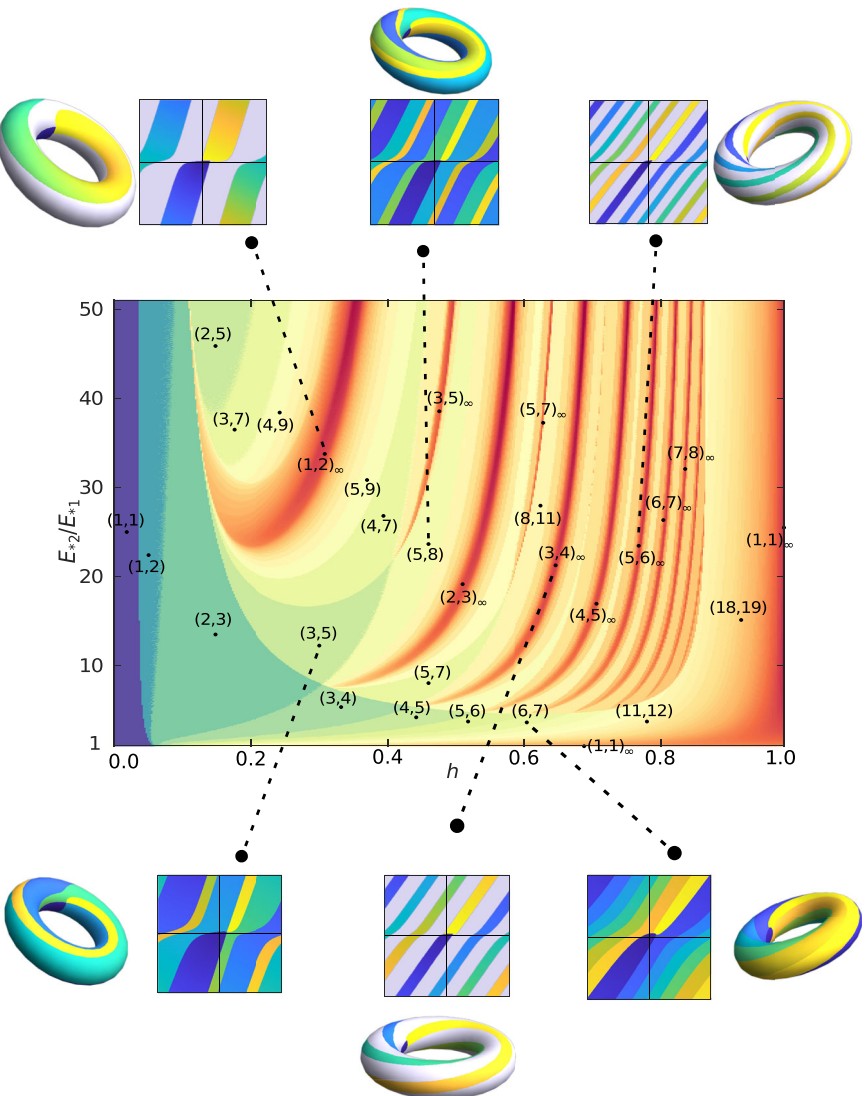

**Fig. 5 | Phase-locking diagram as a function of the coupling strength $h$ and the driving force asymmetry $E_{*2}/E_{*1}$.** The logarithmic colormap indicates the maximum winding of the second oscillator and is used to differentiate the different phase portrait topologies that emerge. The labels indicate the topology in the region marked by the black dots. Selected examples of phase portraits are shown on a two-dimensional projection and on the torus.

increase by $(1, 2)$. With a further increase of $h$, we observe a bifurcation to $(4, 9)$, again by an increment of $(1, 2)$. As we increase $h$ further, we keep undergoing more and more of these bifurcations, effectively climbing up an infinite ladder of the form $(2, 5) + n \times (1, 2)$ with $n = 0, 1, 2, \ldots, \infty$. After only a finite increase in $h$ up to a critical value $h_\infty$, the system has undergone an infinite number of these bifurcations and reaches a TPL state $\lim_{n \to \infty}[(2, 5) + n \times (1, 2)] = (1, 2)_\infty$, i.e., a phase portrait with running band emerges. When $h$ is further increased beyond $h_\infty$, we now descend down a different infinite ladder, out of step with the first one, of the form $(4, 7) + n \times (1, 2)$ with $n = \infty, \ldots, 2, 1, 0$. The system thus ultimately reaches the finite phase locking topology $(4, 7)$. A further increase of $h$ now takes us into the range of influence of the $(3, 5)_\infty$ TPL state, so that the system begins to climb up a new ladder and bifurcates to a $(4, 7) + (3, 5) = (7, 12)$ topology, and so on and so forth. An example of how the phase portraits change as one moves across the $(2, 3)_\infty$ TPL state is shown in Supplementary Fig. 1.

A number of phase portraits for different points on the phase-locking diagram is shown in the insets of Fig. 5, and more examples are shown in Supplementary Figs. 2 and 3. In particular, a number of phase portraits displaying TPL are included. Just like the $(2, 3)_\infty$ trajectory in Fig. 4, periodic trajectories inside these running bands form torus

knots. Such knots are defined by a tuple $(q, p)$ where $q, p$ are coprime to each other and characterize the winding along the two axis of the torus[39]. For all $(m, n)_\infty$ topologies that we have observed, $m$ and $n$ were indeed coprime, suggesting that the TPL states correspond to various torus knots. Torus-knot trajectories have been found in the past in soliton equations, for instance in the non-linear Schrödinger equation[48]. Our system provides a new example of a non-linear dynamical system that can give rise to such mathematical structures.

The phase-locking diagram in Fig. 5 bears some resemblance to the well-known Arnold tongues describing phase locking in a number of other systems[35,49–51]. However, the resemblance is only superficial: in fact, while in the case of Arnold tongues the key parameter controlling phase-locking ratios is the frequency asymmetry and the coupling merely acts to broaden the phase-locking regions, here the opposite is true. The main parameter controlling the phase-locking ratios is the coupling $h$, and the very limited amount of broadening of the phase-locking regions originates from the driving asymmetry $E_{*2}/E_{*1}$. Besides this clear operational difference, the context here is entirely different, as we are still dealing with noise-activated dynamics—although the coexistence of a running band and a stable fixed point leads to a

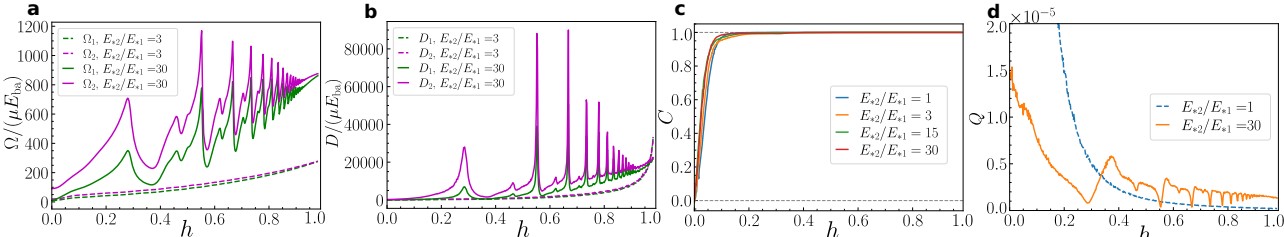

**Fig. 6 | Signatures of TPL states in the stochastic simulations. a** Average speed and **b** diffusion coefficient of the oscillators for weak and strong driving force asymmetry. Peaks are observed as the resonances are crossed for strong asymmetry appearing as a function of the coupling strength. **c** The correlation between oscillators rapidly grows independently of the driving force asymmetry. **d** Quality factor quantifying the stochastic thermodynamics of precision (Eq. (5)). The quality factor drops as the resonances are crossed.

coexistence of dissipative and effectively conservative/deterministic dynamics[18].

Lastly, it is worth commenting on the TPL state $(1, 1)_\infty$, which corresponds to both oscillators advancing equally. Interestingly, this state occurs in two very particular lower dimensional manifolds: on the manifold defined by $h = 1$ (maximum coupling allowed by the positive-definiteness of the mobility matrix), and on the manifold defined by $E_{*2}/E_{*1} = 1$ (the special case of identical oscillators) when $h > h_*$. This latter observation corresponds to the results of ref. [16] and ref. [17], which dealt with identical oscillators and observed $(1, 1)_\infty$ topologies for all values of the coupling above a critical value $h_*$. The existence of TPL over a broad range of values of the coupling strength appears to be a special feature of the symmetric case, as in the general case studied here, we find that TPL states only occur at discrete values of the coupling strength.

For the sake of completeness, we have calculated analogous phase-locking diagrams for other choices of system parameters, see Supplementary Figs. 4 and 5. The overall qualitative features are unchanged.

## Signatures of TPL in the stochastic dynamics

In order to ascertain whether the TPL states in Fig. 5 have an effect on the stochastic dynamics in the presence of noise, we now quantify the long-time behavior of our stochastic simulations. In particular, we will measure the average speed $\Omega_\alpha$ and diffusion coefficient $D_\alpha$ of each oscillator ($\alpha = 1, 2$), and the correlation $C$ between the two oscillators. Defining $\delta\phi_\alpha(\tau; t) \equiv \phi_\alpha(t + \tau) - \phi_\alpha(t)$, we calculate the average speed of oscillator $\alpha$ as

$$\langle \delta\phi_\alpha(\tau; t)\rangle_t \underset{\tau\to\infty}{\sim} \Omega_\alpha \tau, \tag{2}$$

where the operator $\langle \ldots \rangle_t$ denotes a time average over a long simulation. The diffusion coefficient is similarly calculated as

$$\langle [\delta\phi_\alpha(\tau; t) - \langle \delta\phi_\alpha(\tau; t)\rangle_t]^2\rangle_t \underset{\tau\to\infty}{\sim} 2D_\alpha\tau. \tag{3}$$

Finally, the correlation between oscillators is calculated as

$$\frac{\langle \prod_{\alpha=1,2}[\delta\phi_\alpha(\tau; t) - \langle \delta\phi_\alpha(\tau; t)\rangle_t]\rangle_t}{\sqrt{\prod_{\alpha=1,2}\langle [\delta\phi_\alpha(\tau; t) - \langle \delta\phi_\alpha(\tau; t)\rangle_t]^2\rangle_t}} \underset{\tau\to\infty}{\sim} C, \tag{4}$$

and is bounded between −1 and 1 for perfectly anticorrelated and perfectly correlated processes, respectively.

We first considered the average speed $\Omega_\alpha$ as a function of coupling strength for fixed values of the driving asymmetry (Fig. 6a), corresponding to horizontal cuts in the phase-locking diagram of Fig. 5. For small asymmetry, where there are no TPL states, the average speed of both oscillators increases monotonically with increasing coupling. The phenomenology is very different for strong asymmetry, where

increasing the coupling strength takes the system through a series of TPL states. At each of these, we find that the average speed of the coupled slow-fast oscillators sharply peaks. Thus, the presence of a running band strongly enhances the average speed of the oscillators.

An analogous behavior is observed for the diffusion coefficients $D_\alpha$, with a monotonic increase in the absence of TPL states at weak driving force asymmetry, and very sharp peaks when TPL states are crossed at strong driving force asymmetry (Fig. 6b). In analogy with the standard giant diffusion observed for single oscillators at the threshold of noise-activated and deterministic dynamics[40], the giant diffusion for TPL states can be understood as a consequence of the bistability that arises in systems with a running band, which stochastically switch between dissipative dynamics that keep the system at the stable fixed point, and quasi-deterministic dynamics when the system is within the running band[17].

We also note that, independently of the amount of driving force asymmetry, the correlation $C$ quickly grows with increasing coupling, and for $h \gtrsim 0.1$ saturates to $C \approx 1$ indicating perfect correlation between the oscillators, see Fig. 6c. Indeed, from the topology of the deterministic phase portraits, we expect the dynamics of the oscillators to become correlated for any topology other than the trivial topology (1,1), which is present only at very low $h$ independently of the driving force asymmetry.

Lastly, we consider the stochastic thermodynamics of the two coupled processes[52]. In particular, the thermodynamic uncertainty relation (TUR)[4] shows that energy dissipation (or entropy production) puts a fundamental lower bound on the precision of a nonequilibrium process. More precisely, the multidimensional TUR (MTUR) provides the bound $\mathcal{J}^T \mathcal{D}^{-1} \mathcal{J} \le \dot\sigma/k_B$ at steady state, where $\dot\sigma$ is the entropy production rate, $\mathcal{J}$ is any vectorial current, and $\mathcal{D}$ is the diffusion matrix describing the fluctuations of the current[41].

In our two-oscillator system, we have $\mathcal{J}_\alpha = \Omega_\alpha$ and $\mathcal{D}_{\alpha\alpha} = D_\alpha$ for $\alpha = 1, 2$, as well as $\mathcal{D}_{12} = \mathcal{D}_{21} = C\sqrt{D_1 D_2}$. The MTUR can then be rewritten explicitly as

$$Q \equiv \frac{1}{1-C^2}\left(\frac{\Omega_1^2}{D_1} - \frac{2C\Omega_1\Omega_2}{\sqrt{D_1 D_2}} + \frac{\Omega_2^2}{D_2}\right)\frac{k_B}{\dot\sigma} \le 1 \tag{5}$$

where $Q$ is a quality factor, equal to 1 when the bound is saturated (the precision is as high as thermodynamically allowed) and 0 for a purely diffusive process. The entropy production $\dot\sigma$ can be calculated from the steady-state dissipation $T\dot\sigma = F_1\Omega_1 + F_2\Omega_2$.

The behavior of the quality factor $Q$ as a function of the coupling strength $h$ for both weak and strong driving force asymmetries is shown in Fig. 6d. The fact that for strong asymmetry the system crosses through various TPL states with increasing $h$ is clearly signaled in the stochastic thermodynamics of precision. In particular, we see that $Q$ strongly decreases at each TPL state, which may be counter-intuitive considering that the average speed peaks at these states (Fig. 6a). However, note that the diffusion coefficient also strongly

peaks at the TPL states (Fig. 6b), more sharply than the average speed, so that the quality factor ultimately decreases at the TPL states.

## Discussion

Besides the obvious interest from the point of view of dynamical systems theory, we anticipate that our results may find practical applications in a variety of systems. In particular, we have explicitly shown in the Supplemental Methods how Eq. (1) can be derived from a microscopic model of two stochastic rotors that are hydrodynamically coupled (Fig. 1c, d). We previously showed how a dissipative coupling arises when two enzymes that undergo conformational changes during their chemical reactions are in proximity of, or mechanically linked to, each other[16]. In this case, we showed that the coupling constant $h$ becomes phase-dependent, as it depends on the nature of the conformational changes, but the phenomenology remained identical to that observed with constant coupling[16,17].

We hypothesize that the rate enhancements afforded by TPL states could be exploited by enzymes that form heterodimers (that is, complexes of two distinct enzymes) in order to boost the catalytic activity of the slower enzyme. Indeed, some heterodimeric enzymes show higher activity than what could be achieved by the two individual enzymes alone[53,54]. The same kind of rate enhancement could be present in clusters of transmembrane protein channels[20] and rotors[21–24], as well as in groups of kinesins and dyneins walking on the same microtubule[55], or different myosins exerting contractile forces on nearby actin filaments[56]. Alternatively, TPL states may be targeted in engineered systems whose dissipative coupling and driving force asymmetry can be experimentally controlled, such as superconducting Josephson junction arrays[18,27], laser cavities[31,32] or optomechanical devices[33,34].

Future work could explore the dissipatively-coupled dynamics of many ($N > 2$) nonidentical oscillators, which would allow for the study of e.g., disorder in the microscopic properties of the various oscillators, or of the interactions between groups of oscillators. In the case of identical oscillators, we previously found that for large $N$, noise activation becomes irrelevant and the dynamics become effectively deterministic, while the correlations between oscillators vanish[17]. Other interesting modifications to the dynamics studied here could include the effect of time delays or memory in the dissipative coupling, representing e.g. mechanical interactions between oscillators mediated by a viscolastic medium; as well as the breakdown of the fluctuation-dissipation relation, which would make the mobility matrix that enters the deterministic dynamics distinct from the diffusion matrix that enters the stochastic dynamics. Indeed, previous work has shown that a strongly correlated noise, even in the absence of a deterministic coupling, is enough to induce synchronization in excitable systems[57].

## Methods

### Stochastic simulations

To integrate the stochastic differential equations, Eq. (1), we employed the Euler−Maruyama method using a custom code written in the Julia language[58]. Time was nondimensionalized as $\tilde{t} = \mu_1 v_1 t$. For the results in Fig. 6, a time step $d\tilde{t} = 10^{-2}$ was used, with the total number of steps equal to $10^9$ and the number of samples equal to $10^6$. We also averaged over ten different runs. More detailed information on how the observables in Eqs. (2)–(4) were calculated and can be found in the supplementary information of ref. 17.

### Phase portraits

To generate phase portraits, we integrated the deterministic equations of motion (corresponding to Eq. (1) without the noise term) using the built-in *ode45* integrator in MATLAB[59], which employs a 4th-order Runge-Kutta method. A 301 × 301 grid of initial points in the interval $-\pi < \phi_{1,2} < \pi$ was used, and we integrated the trajectories up to a

maximum integration time $\tilde{t}_{\max} = 100$. The final points were then used to identify the winding number $(m, n)$ if the trajectory reached a stable fixed point, or $(m, n)_\infty$ if the trajectory was found to be periodic and thus to lie on a running band.

## Data availability

The data supporting the findings of this study are available in the paper and its Supplementary Information.

## Code availability

The algorithms for the codes supporting the findings of this study are available in the paper and its Supplementary Information.

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

## Acknowledgements
We acknowledge support from the Max Planck School Matter to Life and the MaxSynBio Consortium which are jointly funded by the Federal Ministry of Education and Research (BMBF) of Germany and the Max Planck Society.

## Author contributions
M.C., R.G., and J.A.-C. designed the research, conducted the research, analyzed the data, and wrote the paper.

## Funding

## Competing interests
The authors declare no competing interests.
