## [Transparent Peer Review file · Nature Communications]

Topological phase locking in stochastic oscillators

Corresponding Author: Dr Jaime Agudo-Canalejo

Version 0:

Reviewer comments:

Reviewer #1

(Remarks to the Author)

The work by Michalis et al. presents an anomalous stochastic dynamic (TPL: topological phase locking) in two noise-activated oscillators with dissipative coupling. By adjusting the driving force asymmetry E^2/E^1 and the coupling strength h in the Langevin equation, their numerical results show the phase trajectory from non-closed to closed in parameter space (ϕ_1, ϕ_2). The winding number pair is fixed for the closed trajectory; hence, it is called TPL. They attribute TPL to a bifurcated structure and find that TPL states can enhance the average speed of oscillators. Although the methods used in this paper have been developed in their previous work [Phys. Rev. Lett. 127, 208103 (2021); New J. Phys. 25, 093014 (2023)], the TPL is novel and interesting, and the paper is well written. There are some concerns that need to be addressed.

Usually, winding numbers or other topological invariants are defined in a close region. Why are there winding number pairs for the non-closed trajectory in Fig 3?

Connecting biological phenomena and physical or mechanical processes is a good idea. However, my main comment is that the research is merely mathematical. In fact, except for the background section and the outlook part, there is no connection to enzymes or molecules in the whole text. So, there is no biological evidence to support TPL in molecular oscillators. Science should be cautious about what is uncertain. The current data does not support its title unless it results from a molecular experiment.

In general, TPL is interesting, but the current version doesn't support its strong correlation with biology. I suggest combining this phenomenon with achievable physical systems, which would also be an excellent job.

Reviewer #3

(Remarks to the Author)

Through the manuscript titled "Topological phase locking in molecular oscillators", the authors have identified a novel phenomenon called topological phase locking by examining a thermodynamically consistent model of two interconnected noise-driven oscillators. This mechanism leads to quasi-deterministic dynamics, significantly boosting the average oscillation speed and giant diffusion. Topological phase locking manifests as a set of periodic orbits that form intricate torus knots in phase space, where the oscillators move in rational ratios relative to one another. This phenomenon contrasts with the typically random behavior of molecular enzymes and motors, which operate out of equilibrium under the influence of thermal noise, with energy dissipation on a scale comparable to thermal energy. The effectively conservative dynamics due to topological phase locking coexist with a dissipative fixed point, creating a dual dynamic regime. The authors further demonstrate that this dynamics emerges through a complex hierarchy of global bifurcations.

I personally found the study to be engaging, with notably novel results and a commendable presentation and approach. The results presented in this study are significant, and the authors have conducted a thorough analysis that, in my opinion, aligns well with the aims and scope of the journal Nature Communications.

I do not have major concerns with the current version of the manuscript. Therefore, the article may be considered for acceptance after it addresses the following comments:

1) I am curious about how the observed phenomenon extends to larger networks beyond just two coupled systems.

Including a discussion on this aspect would be valuable and insightful for interested readers.

2) I was wondering about the potential effects of time-delayed interactions on the phase-locking scenario explored in this study. A discussion on this could provide additional depth and understanding.

Reviewer #4

(Remarks to the Author)

Topological phase locking in molecular oscillators"

by Michalis Chatzittofi, Ramin Golestanian and Jaime Agudo-Canalejo
The Authors report on a new phenomenon in molecular oscillators called Topological Phase Locking (TPL). Convincing arguments on the deterministic dynamics of TPL on tori are offered. These new results however seem to emerge from studies recently published by the same authors (see Ref. 16, M Chatzittofi et al 2023 New J. Phys. 25 093014).

So although the results are worthy of publication I am asking the Authors to stress the differences with respect to their Ref.16.

There is another aspect that I would like to report to the Authors regarding another work (not cited by the authors) where the synchronization between two driven FitzHugh-Nagumo oscillators coupled by white and colored noise is reported experimentally ("Synchronization of uncoupled excitable systems induced by white and coloured noise" New Journal of Physics 12 (2010) 053040. It is easy to understand that if the driving were not at the same frequency the synchronization would occur on a torus.

In conclusion, the manuscript is susceptible to publication but precise comments on previously published papers by the same Authors and one on an excitable system with notable impact in neural dynamics must be introduced.

Version 1:

Reviewer comments:

Reviewer #1

(Remarks to the Author)

I understand that the "strong connection" mentioned by the author originates from mathematical evolution equations describing biological enzymes or molecular motion. However, as I previously noted, the series of derived results in this paper (including "topological phase locking") are purely mathematical. If these results explicitly predict or explain phenomena related to biological enzymes or molecular motion, I would fully endorse the authors' claims. If not, the authors can only propose that topological phase locking in random oscillators might (note the emphasis on "might") be potentially related to certain biological phenomena.

As the authors mentioned in their response that they do not wish to overstate the biological significance of their findings, they must explicitly present biological evidence supporting their conclusions. If such evidence were provided in the paper, I would unhesitatingly recommend acceptance by Nature Communications. However, I believe the current revised version would be more appropriately suited for publication in a more specialized journal.

Reviewer #3

(Remarks to the Author)

In the revised version of the manuscript, the authors have addressed all of my previous concerns satisfactorily. I therefore recommend that the revised manuscript may now be accepted for publication.

Reviewer #4

(Remarks to the Author)

The manuscript has improved considerably taking into account the criticisms of the three referees. As far as I am concerned the work can be published.

Topological phase locking in stochastic oscillators
Manuscript reference: NCOMMS-24-26587-T
Response to the comments of the referees
(Dated: April 8, 2025)

This document contains our detailed answers to the comments of Referees. Page and reference numbers mentioned in our answers refer to the updated and marked version of the manuscript sent in this resubmission. In the updated and marked version, blue pieces of text correspond to pieces added or edited in response to the referees' comments. Below, the comments of the referees are displayed in *italics*, while our answers are in the standard font.

REFEREE 1

1. *Usually, winding numbers or other topological invariants are defined in a close region. Why are there winding number pairs for the non-closed trajectory in Fig 3?*

Indeed, winding numbers are usually defined for closed trajectories. In our convention, pairs of winding numbers for closed trajectories (and the associated phase portrait topologies) are denoted as having a ∞ subscript, i.e. $(m, n)_\infty$ as in Fig. 4. Nevertheless, to classify non-closed trajectories that end at the stable fixed point as in Fig. 3, we also assign them “winding numbers” following an unambiguous procedure: a trajectory starting anywhere in the region $-\pi \leq \phi_1, \phi_2 < \pi$ is said to have winding number (m, n) if it ends at the stable fixed point at $(\phi_1, \phi_2) = (2\pi m, 2\pi n)$. This allows us to differentiate the basins of attraction in the region $-\pi \leq \phi_1, \phi_2 < \pi$, and to classify different phase portrait topologies according to the highest winding number (m, n) that appears on it.

Our choice of notation is motivated by the connection that we find between phase portraits with open and closed trajectories, described in pages 5-6. The phase portraits with closed trajectories $(m, n)_\infty$ appear as the limit of phase portraits with open trajectories (a, b) after an infinite number of bifurcations that increase their winding number by (m, n) . In mathematical notation, we find that $\lim_{k \rightarrow \infty} [(a, b) + k \times (m, n)] = (m, n)_\infty$.

In the revised version, we have further clarified our notation with new pieces of text on pages 4 and 5.

2. *Connecting biological phenomena and physical or mechanical processes is a good idea. However, my main comment is that the research is merely mathematical. In fact, except for the background section and the outlook part, there is no connection to enzymes or molecules in the whole text. So, there is no biological evidence to support TPL in molecular oscillators. Science should be cautious about what is uncertain. The current data does not support its title unless it results from a molecular experiment.*

We thank the referee for raising this concern. Since the beginning, our motivation for this research came from thinking about enzymes and molecular motors, and we let this motivation slide into our title. We note that, besides the background section and the outlook part, we also make a strong connection to molecular oscillators in the Supplementary Information, where we explicitly derive Eq. (1) in the main text from a microscopic model of molecular rotors. In particular, we give explicit forms for μ_1 , μ_2 , and h that define the matrix M_{ij} in Eq. (1) in terms of the parameters of the microscopic model.

In any case, we agree with the referee that our work is more generally about stochastic oscillators, not necessarily biomolecular systems. We neither want to overstate the biological implications of our results, or to understate their generality. We have therefore changed the title to “Topological phase locking in stochastic oscillators” in the revised version.

REFEREE 2

1. *I am curious about how the observed phenomenon extends to larger networks beyond just two coupled systems. Including a discussion on this aspect would be valuable and insightful for interested readers.*

We thank the referee for raising this point. We agree that it would be very interesting to investigate this. We note that in [Chatzittofi, M., Golestanian, R., & Agudo-Canalejo, J. (2023). Collective synchronization of dissipatively-coupled noise-activated processes. *New Journal of Physics*, 25, 093014.] we already looked at the case of N identical oscillators. Future work could explore N non-identical oscillators, either with parameters

randomly selected from a distribution (allowing us to explore the effects of disorder), or by splitting the N oscillators into subgroups with different parameters for each subgroup. In the revised version, we now discuss potential avenues for future exploration in the Discussion section.

2. *I was wondering about the potential effects of time-delayed interactions on the phase-locking scenario explored in this study. A discussion on this could provide additional depth and understanding.*

We also thank the referee for this suggestion. It is hard to speculate about what the outcome of time-delayed interactions will be in this system without explicitly studying it, which would be beyond the scope of the present work. We do think however that this is a very promising direction for future work, as it could represent e.g. mechanical interactions mediated by a viscoelastic medium. In the revised version, we now mention this in the Discussion section.

REFEREE 3

1. *I am asking the Authors to stress the differences with respect to their Ref.16.*

We would like to thank the referee for this comment. The main differences between the present manuscript and Ref. 16 is that in that reference we were dealing with N identical oscillators, while here we are dealing with two *non-identical* oscillators. Crucially, all the rich phenomenology that we discover in the present manuscript hinges on the two oscillators being non-identical. In the notation of the present manuscript, the case of identical oscillators only shows the $(1, 1)$ and $(1, 1)_\infty$ phase portrait topologies (see horizontal axis $E_{*2}/E_{*1} = 1$ in Fig. 5). The wealth of new phase portrait topologies and associated TPL “resonances” (Fig. 5) that we observe is only possible thanks to the two oscillators being non-identical.

In the revised version, we now stress further the differences between the two cases on pages 1, 3, and 7.

2. *There is another aspect that I would like to report to the Authors regarding another work (not cited by the authors) where the synchronization between two driven FitzHugh-Nagumo oscillators coupled by white and colored noise is reported experimentally (“Synchronization of uncoupled excitable systems induced by white and coloured noise” *New Journal of Physics* 12 (2010) 053040. It is easy to understand that if the driving were not at the same frequency the synchronization would occur on a torus.*

We thank the referee for pointing us to this interesting paper. That paper explores the synchronization of uncoupled excitable systems due to a common noise. One similarity with our work is that, in our case, the off-diagonal coefficients of the mobility matrix similarly induce correlations in the noise experienced by the two oscillators. However, in our case these off-diagonal coefficients also induce a coupling at the deterministic level.

In the revised version, in the Discussion section we now mention the work suggested by the referee in the context of a breakdown of the fluctuation-dissipation relation.

NCOMMS-24-26587B: Response to comments of the reviewers

Reviewer #1:

We thank the reviewer for their comments. As the reviewer says in their review, we “only propose that topological phase locking in random oscillators might (note the emphasis on "might") be potentially related to certain biological phenomena.” We do not make bigger claims than that. We appreciate that the reviewer thinks this makes our work more suitable for a specialized journal but we think that, even beyond the direct or only speculative relation to biological phenomena, our work is of sufficient importance within physics and dynamical systems theory to merit publication in Nature Communications.

Reviewers #3 and #4:

We thank the reviewers for their consideration of our manuscript and their recommendation of acceptance.